# Conserved wing shape variation across biological scales unveils dialectical relationships between micro- and macroevolution
Keita Saito[1], Masahito Tsuboi [2,4] ✉ & Yuma Takahashi [3,4] ✉

Variation enables short-term evolution (microevolution), but its role in long-term evolution (macroevolution) is debated. Here, we analyzed a dataset of *Drosophila* wing variation across six levels of biological organization to demonstrate that microevolutionary variation and macroevolutionary divergence are positively correlated at all levels from variation within an individual to macroevolution over 40 million years. Surprisingly, the strongest relationship was between developmental noise and macroevolutionary divergence—which are traditionally considered the most distant— while the relationship between standing genetic variation and population divergence was modest, despite established theoretical predictions and empirical evidence. Our results indicate that the congruence of the developmental system with the long-term history of fluctuations in adaptive peaks creates dialectical relationships between microevolution and macroevolution.

Since the modern synthesis of evolutionary biology, it has been argued that microevolutionary processes of mutation, selection, genetic drift, and gene flow operating within populations can explain, or at least are consistent with, evolution at long timescales[1]. Although debates on the role of additional processes operating above the population level remain[2,3], the extrapolating view of macroevolution profoundly influences our current thinking of evolution[4]. Arguments in favor of this view rest on evidence from evolutionary genetics. Previous studies have demonstrated that the pattern of phenotypic divergence in high-dimensional phenotype space is biased in directions that harbor high amounts of additive genetic variance[5–7] (e.g., the genetic line of least resistance). Rapidly accumulating evidence indicates that macroevolutionary divergence in various traits and taxa is predictable from standing genetic variation in contemporary populations[8–11], as predicted if genetic constraints are relevant for macroevolution. The increasing evidence for the constraints hypothesis provides apparent support for the extrapolating view of macroevolution, but the mechanisms underlying this inference remain elusive (e.g., the paradox of predictability)[12].

The paradox of predictability could be reconciled if variability—the ability of a developmental system to produce variation[13]—mirrors the pattern of long-term fluctuations in adaptive peaks. This idea, hereafter referred to as the congruence hypothesis, challenges the established theory of evolutionary genetics by arguing that the standing genetic variation of a population (i.e., evolvability)[14] is a consequence, rather than a cause, of macroevolution. Two lines of evidence support this hypothesis. First, it has been shown that the pattern of phenotypic plasticity is consistent with a major axis of phenotypic divergence through computer simulations[15], laboratory experiments[16], and meta-analyses[17]. Second, Rohner and Berger[18] have recently demonstrated that the subtle responses of development to random and localized environmental fluctuations (i.e., developmental noise)[19,20] are correlated with the pattern of macroevolutionary divergence. The congruence hypothesis offers a new class of mechanisms regarding the role of non-genetic variations in evolution[21–24], in ways that are consistent with the established theory of genetics[25,26].

In evaluating the constraints and congruence hypotheses, measures of variance at different levels of biological organization are necessary. Here, a multivariate approach was used to estimate the pattern of variation in the wing morphology of *Drosophila* flies at six levels: (i) species divergence, (ii) population divergence within a species (*D. simulans*), (iii) genetic variation within a population, (iv) phenotypic plasticity, (v) mutational variance, and (vi) developmental noise. Each variation was summarized as the co/variance matrix and expressed as follows: **R**, divergence among 112 species; $\mathbf{D}_{sp}$, divergence among 14 species; $\mathbf{D}_{pop}$, population divergence; **G**, genetic variation; **E**, phenotypic plasticity; **M**, mutational variance; **F**, developmental noise.

[1]Graduate School of Science and Engineering, Chiba University, Chiba, Japan. [2]Department of Biology, Lund University, Lund, Sweden. [3]Graduate School of Science, Chiba University, Chiba, Japan. [4]These authors contributed equally: Masahito Tsuboi, Yuma Takahashi. ✉e-mail: masa.tsuboi@gmail.com; takahashi.yum@gmail.com

**Table 1 | List of used species for estimating matrix and the way of estimating**

| Matrix | Used Species | Estimating way | Sample size |
|--------|--------------|----------------|-------------|
| **R** | 112 species | From the previous study[c] | 112 species |
| $\mathbf{D}_{sp}$ | 14 species[a] | Our MCMCglmm model | 14 species |
| $\mathbf{D}_{pop}$ | *D. simulans* | Our MCMCglmm model | 11 populations |
| **G** | *D. simulans* | Our MCMCglmm model | 33 isofemale lines |
| **E** | *D. simulans*[b] | Our MCMCglmm model | 8 environments |
| **F** | *D. simulans* | Our MCMCglmm model | 7654 wings[¶] |
| **M** | *D. melanogaster* | From the previous study[d] | 12,075 wings |

It includes both left and right wings and replicated measurements.
[a]It includes the wing photos obtained from DrosoWing Project.
[b]It includes the wing photos obtained from Saito et al. (2024).
[c]Houle et al. (2017).
[d]Houle and Fierst (2013) and Houle et al. (2017).

We will examine the relationship of variation across these six levels and discuss the role of constraints and congruence hypotheses to explain the paradox of predictability using theoretical predictions of quantitative genetics[27,28] and statistical physics[29] as our guiding principles. Our basic predictions are that the relationship between standing genetic variation (**G**) and microevolution ($\mathbf{D}_{pop}$) will support the constraints hypothesis, while the relationship between variability (**F** or **M**) and macroevolution (**R**) will support the congruence hypothesis. By examining the degree of support for these predictions based on the exponent (i.e., the slope of log-log regression) and the coefficient of determination ($R^2$) of the relationships, we will propose a pluralistic perspective for understanding the relationship between microevolution and macroevolution.

## Results and discussion
### Correlation between mutational variance and fluctuating asymmetry

Two theoretical frameworks have been proposed to elucidate the relationship between variation and divergence. First, the quantitative genetic model of Lande[27] proposed the following relationship:

$$\Delta \bar{z} = \mathbf{G}\beta$$

where $z$ denotes a phenotype, **G** represents the additive genetic co/variance matrix, and $\beta$ is the selection gradient (the covariance between phenotype and relative fitness). Under the assumption that genetic drift is the sole driver of evolutionary changes, this framework predicts a positive relationship between **G** and divergence among conspecific populations, with the scaling exponent of one. If **G** remains stable for a long period of time and the drift dictates evolution, then the pattern of genetic constraints characterized by **G** scales up to the divergence across species and higher taxa[1]. The influence of directional, disruptive, and stabilizing selection on the relationship between variation and divergence can be described by:

$$\mathbf{D} = \mathbf{G}\gamma\mathbf{G}$$

where **D** represents phenotypic divergence based on the mean trait values among populations within a species and $\gamma$ is the multivariate selection matrix[30,31]. When selection influences the pattern of divergence, the exponent of the relationship between variation and divergence is expected to deviate from 1. Specifically, the exponent will range from 0 when divergence is primarily driven by rapid fluctuations in adaptive optima, to 2 when directional selection dominates the divergence process[8,11].

The second framework, based on the quantitative genetic model of Lynch and Hill[28] and the statistical physics model of Kaneko and Furusawa[29], indicates that variability is proportional to the pattern of long-term evolutionary divergence among species. Traditionally, variability has been measured as the subset of phenotypic variance attributable to spontaneous mutation, which is summarized in the mutational co/variance matrix, **M**[32]. Based on **M**, Lynch and Hill[28] proposed that the rate of macroevolution (**R**) is proportional to 2 **M**:

$$\mathbf{R} \propto 2\mathbf{M}$$

In a related model, Kaneko and Furusawa[29] conceptualized variability as developmental fluctuation (developmental noise), and they proposed that the rate of evolution is proportional to the variance of developmental noise $\langle(\delta X)^2\rangle$. This evolutionary fluctuation–response relationship can be expressed as follows:

$$\frac{\langle X\rangle_{a+\Delta a} - \langle X\rangle_a}{\Delta a} \propto \langle(\delta X)^2\rangle$$

where $\langle X\rangle_a$ and $\langle(\delta X)^2\rangle = \langle(X - \langle X\rangle)^2\rangle_a$ represent the average and variability, respectively, of the phenotypic trait $X$ for a given system parameterized by $a$. The relationship described above was derived under the assumption that the distribution $P(X; a)$ follows an approximately Gaussian form, with the effect of changes in $a$ on the distribution represented by a bilinear coupling between $X$ and $a$. When $a$ is assigned as a parameter that specifies the genotype, the left-hand side of the equation, $(\langle X\rangle_{a+\Delta a} - \langle X\rangle_a)/\Delta a$, quantifies the phenotypic change resulting from a genetic alteration and represents the rate of evolution (**R**)[33]. Therefore, the above relationship can be reinterpreted as a proportional relationship between phenotypic fluctuations due to developmental noise and the rate of evolution. Both models predict positive correlations between variability and the rate of macroevolution with the scaling exponent of one.

The two formulations differ in two key aspects. First, while Lynch and Hill[28] consider **M**—a property of the organism—as a model parameter, Kaneko and Furusawa[29] treat $a$—a property of the system in which the organism resides—as a model parameter. Second, the two models evaluate variability differently. **M** is estimated from mutation accumulation experiments[34], whereas developmental noise can be measured as the random difference between the left- and right sides of laterally symmetric homologs (fluctuating asymmetry, FA)[35–37]. FA represents several favorable attributes as a measurement of variability[18], but to operationalize FA, and therefore the formulation of Kaneko and Furusawa[29] in the context of genetics, we need to establish that FA is correlated with heritable variability (i.e., **M**).

Thus, we compared **M** in *D. melanogaster*, estimated by Houle and Fierst[38] and Houle et al.[8] and the co/variance matrix that represents the variation caused by FA (**F**) in *D. simulans* (Table 1). Two types of **M** were used: spontaneous mutational variance measured under homozygous ($\mathbf{M}_{hom}$) and heterozygous ($\mathbf{M}_{het}$) conditions. **F** was estimated using a random mixed-effect model with repeated measurements from one population of *D. simulans*. The rationale for not estimating **M** in *D. simulans* and for using two types of **M** is provided in the Supplementary Note 1. Linear regression analysis of **M** against **F** revealed a strong positive relationship (Fig. 1; $\mathbf{M}_{hom}$ on **F**: $R^2 = 0.83$, $\beta = 0.67 \pm 0.08$; $\mathbf{M}_{het}$ on **F**: $R^2 = 0.95$, $\beta = 0.73 \pm 0.07$). Therefore, the two kinds of variabilities considered by Lynch and Hill[28] and Kaneko and Furusawa[29] may be interpreted as a measure of the property of a developmental system that translates external perturbations or inputs, such as genotype and environmental parameters, into phenotypic outcomes[39,40]. Given that genotype and environmental parameters are fixed when measuring **M** and **F**, **M** captures the effect of de novo mutation on phenotype, whereas **F** captures the effect of intrinsic developmental perturbation on phenotype. Consequently, both matrices could measure the robustness of phenotype against different sources of perturbations. Based on the framework developed by Lynch and Hill[28] and Kaneko and Furusawa[29], the robustness of the phenotype during developmental process could affect the rate of evolution and serve as a primary source of constraint in evolution.

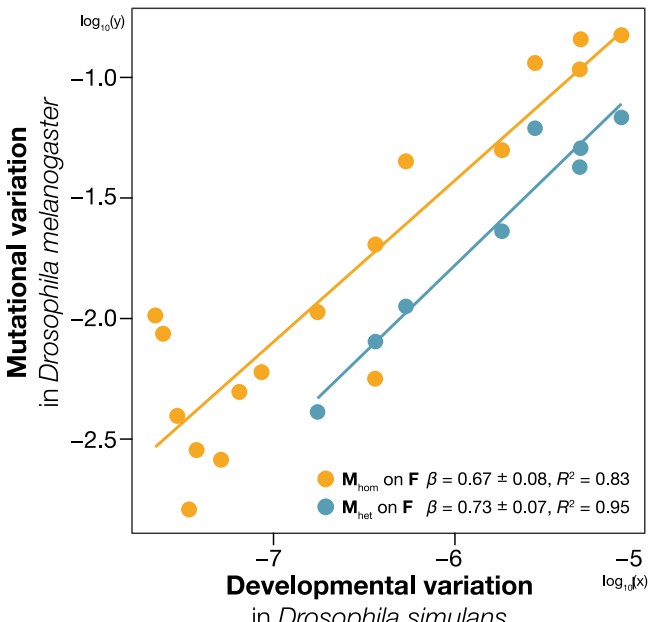

**Fig. 1 | Relationship between two kinds of variabilities.** Points represent $\log_{10}$ (variance in **M** or **F**) along the eigenvectors of **G** in *Drosophila melanogaster*. Key gives log–log regression result, $\beta \pm$ s.e. and $R^2$. Lines indicate ordinary least squares regression lines, and they have slopes equal to $\beta$. Only upper 17 dimensions of $\mathbf{M}_{hom}$ and eight dimensions of $\mathbf{M}_{het}$ were used.

## Variances are positively correlated with divergence at all levels

Next, we examined the validity of the constraints hypothesis[27] and the congruence hypothesis[28,29] by comparing the scaling exponent (i.e., the slope of a log–log regression) and the coefficient of determination ($R^2$) between divergence and variation or variability. A series of experiments was conducted to estimate the standing variation caused by heritable variation (**G**) and plasticity (**E**) of *D. simulans*. Divergence was evaluated at three levels: divergence among populations of *D. simulans* ($\mathbf{D}_{pop}$), divergence across 14 *Drosophila* species ($\mathbf{D}_{sp}$) from an online database (DrosoWing Project)[41] and our dataset, which includes the images used to estimate **G**, **E**, and $\mathbf{D}_{pop}$, as well as wing photos of *D. lutescens*, and divergence across 112 species of Drosophilidae representing 40 million years of evolution (**R**) from Houle et al.[8] (Table 1). If the genetic constraints underpin the relationship between microevolution and macroevolution, then the strongest relationship (i.e., high $R^2$) and the scaling exponent of 1 are predicted between **G** and $\mathbf{D}_{pop}$. Conversely, if the congruence hypothesis underlies the micro–macro relationships, then the strongest relationship is predicted between **F** and **R** with the scaling exponent of <1 because the influence of directional selection is expected to be low at long evolutionary timescales. In addition, we predict that the $R^2$ will be high when the relationship captures a biologically causal link, while such causal relationship should become progressively weaker (i.e., $R^2$ decreases) as the cause and effect will become further and further apart because measurement and estimation errors are added at each level.

A correlation matrix of the relationships across all pair-wise relationships among variation, variability, and divergence revealed universally positive relationships between the three descriptors of variation (genetic variance, plasticity, and variability) and three levels of divergence (Fig. 2). In addition, to avoid the potential impact of using different species' **M**, we conducted additional comparisons between **M** and the other matrices (Supplementary Fig. 1). The correlations are all high (Supplementary Table 1), which indicates that the variation in the 20-dimensional phenotype space of the *Drosophila* wing morphology is packed into a low-dimensional manifold in a remarkably similar manner, regardless of the causes of variation or the levels of divergence. These similarities are evident in the relative change in the position of each landmark on the wing (Fig. 3). Most landmarks showed a propensity to vary along the lateral axis,

particularly landmarks 7, 8, 9, and 10, while the landmarks at the base of the wing exhibited relatively low variation.

The comparison of the coefficient of determination ($R^2$) and slope ($\beta$) showed that the relationship between **F** and **R** is the strongest ($R^2 = 0.85$, $\beta = 0.67 \pm 0.07$) among all pair-wise relationships. This result is surprising because these two levels represent the two most distant levels in the biological hierarchy, namely, internal subtle variability of an individual (**F**) and 40 million years of *Drosophila* wing evolution (**R**). Equally surprising is that the relationship between **G** and $\mathbf{D}_{pop}$ is modest ($R^2 = 0.58$, $\beta = 0.80 \pm 0.16$), although these two levels are thought to be adjacent to each other and have established theoretical and empirical bases to predict a positive relationship[6,27]. These observations support the congruence hypothesis as a leading explanation for the relationship between variation and divergence in phenotypic evolution[8–11].

## Congruence of variability with macroevolution underlies the pattern of *Drosophila* wing-shape evolution

The results shown in Fig. 2 are based on the posterior mode of the co/variance matrices estimated at the respective levels (except for **R**). However, considering the difficulty of estimating variation and variability with high precision, some of the results may reflect estimation errors. To examine the impact of the estimation accuracy of each matrix on our inference, the posterior distribution obtained from the Bayesian mixed model was analyzed. The ordinary least-squares regressions were used with the co/variance matrices describing each level of divergence presented in Fig. 2 as the response variable and one of the 1000 posteriors of **G**, **E**, or **F** as the explanatory variable to obtain the distribution of the coefficient of determination ($R^2$) and the scaling exponent (log–log slope, $\beta$) of the relationships. This analysis (Fig. 4) confirmed that the relationship between **F** and **R** had the highest $R^2$ value among all pairs, which is significantly higher than the $R^2$ value between **G** and $\mathbf{D}_{pop}$.

From a purely statistical point of view, **R** and **F** are the most reliably estimated matrices in our dataset, and this clearly has contributed to a tight relationship between these two matrices. **R** is estimated from 21,138 wings representing 117 taxa using a phylogenetic mixed model to extract phylogenetic variances[8]. **F** is measured by controlling many sources of variation and instrumental measurement error to extract variances only attributable to local environmental perturbations and is estimated from 7654 wings in a single species (Table 1). By contrast, other matrices (i.e., $\mathbf{D}_{sp}$, $\mathbf{D}_{pop}$, **G**, and **E**) are subject to substantial errors that should have contributed to the low $R^2$ value. For example, (i) $\mathbf{D}_{sp}$ is estimated on the basis of the divergence among 14 species; (ii) **G**, **E**, and $\mathbf{D}_{pop}$ are measured at a certain time from a few representative lines of a single species; (iii) **G** is known as a "broad-sense" **G** that confounds additive with non-additive effects (e.g., epistasis and dominance), and (iv) **E** is at best a poor representation of the inherent plasticity because the environmental conditions that are tolerable by the reared genotypes of *D. simulans* in our experiments are likely a small subset of the range of plasticity available in wild populations of this species[42]. Given these substantial errors, it is remarkable that all matrices are still unambiguously correlated. Hence, we interpret our results to suggest that all matrices studied here are biologically connected to one another. Rather than placing emphasis on any one of the levels of biological organization as the cause of universal correlations, we favor a "dialectical" perspective (*sensu* Levins and Lewontin[43]). According to Levins and Lewontin[43], a dialectical perspective holds that the whole and its parts are interdependent, with no predetermined directions of causality across levels of biological organization. In the context of our explanation of the paradox of predictability, a dialectical framework would oppose privileging any single level—whether it is mutational variance (**M**), developmental noise (**F**), standing genetic variation (**G**), plasticity (**E**), microevolution ($\mathbf{D}_{pop}$) or macroevolution (**R**)—as the primary explanatory factor. Instead, all levels are considered interconnected parts of a dynamical system that co-evolve through complex causal relationships[44,45], where multiple directions of causes and effects play distinct and complementary roles. Three observations support our view.

**Fig. 2 | Relationships between variations (G, E, and F) and divergence ($D_{pop}$, $D_{sp}$, and R).** Points represent $\log_{10}$ (variance in each matrix) along the eigenvectors of **G** in *Drosophila melanogaster*. Key gives log–log regression result, $\beta \pm$ s.e. and $R^2$.

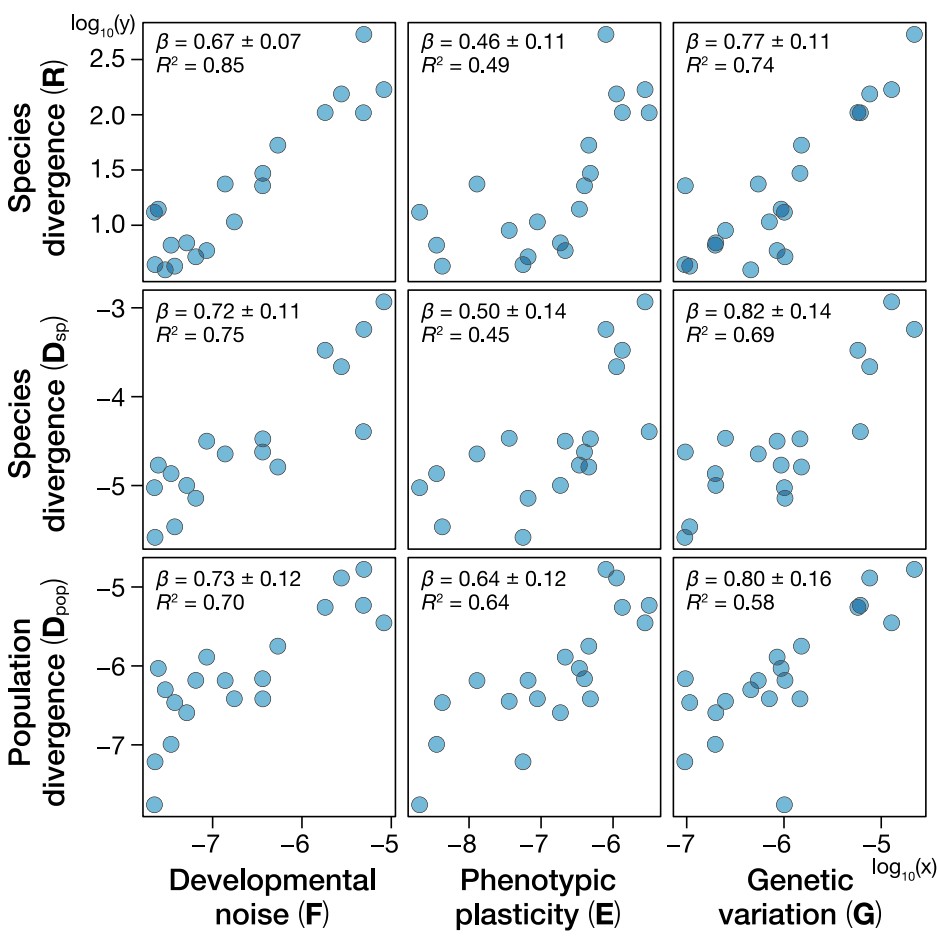

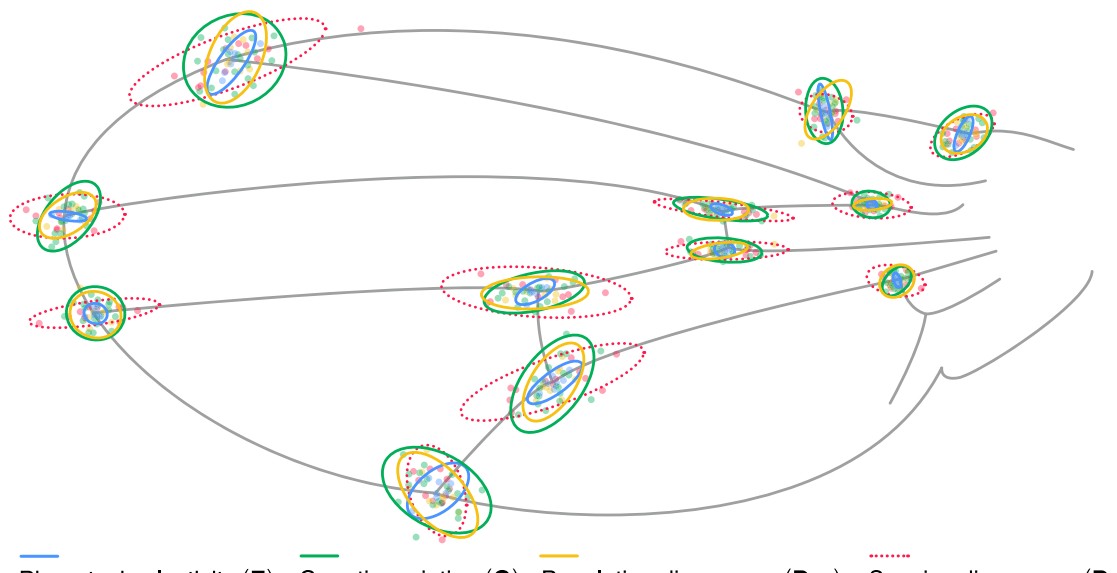

**Fig. 3 | Ellipses representing variation in landmark position.** Ellipses are centered, and they represent six SDs (only $D_{sp}$: 1.5 SD) for better visibility. Points represent the mean values of species ($D_{sp}$), populations ($D_{pop}$), genotypes (**G**), and rearing environments (**E**).

First, the tight relationship between **R** and **F** indicates that the macroevolutionary history of fluctuations in adaptive peaks has shaped the developmental system of the *Drosophila* wing. Our causal hypothesis is explicitly that macroevolution determines variability and not vice versa, because if **F** were the cause and **R** were the effect, the measurement and estimation errors induced at each level (**F** or **M** → **G** or **E**, **G** or **E** → $D_{pop}$, $D_{pop}$ → **R**) would have progressively weakened the relationship. Moreover, the evolution of co/variance matrices should have decayed the ability of our estimates obtained from a single species (*D. simulans*) to predict evolution when wider taxonomic groups are considered[6], but our results revealed the

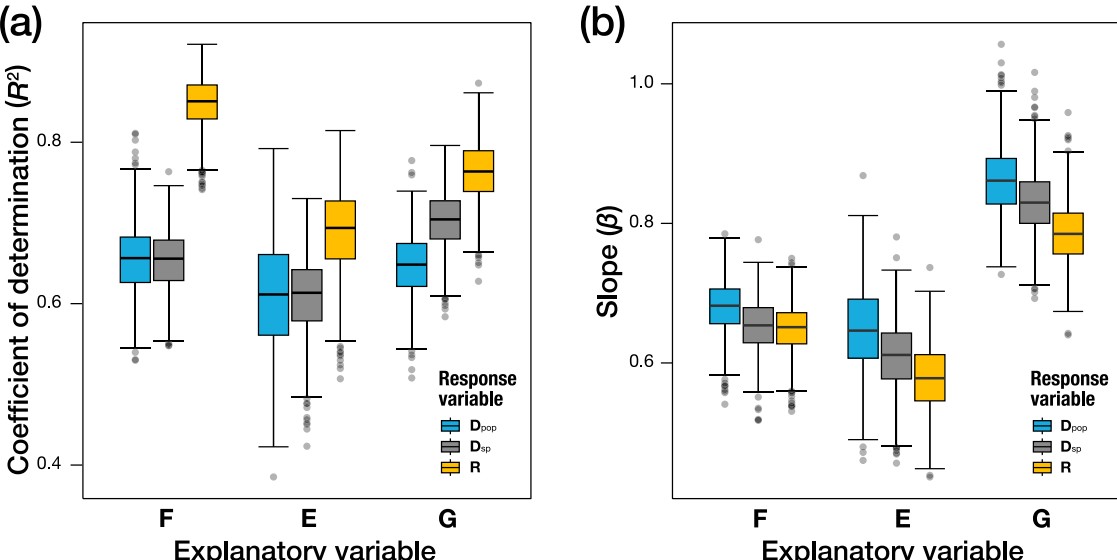

**Fig. 4 | Distributions of $R^2$ and slope in each regression with variation as explanatory variables and divergence as response variables.** Color difference represents the difference in response variables. **a** Distributions of $R^2$ and **b** distributions of slopes. Each boxplot was constructed from I = 1000 data points.

opposite. The congruence between macroevolutionary variation and the property of organisms (developmental and genetic systems) is relevant to macroevolutionary models of trait evolution proposed by Simpson[46] and Hansen[47], which suggest that macroevolutionary phenomena can be interpreted as the average effect of patterns and processes observed along individual lineages. The evidence supporting the tight link between **F** and **R** lends credibility to the idea[46,47] that macroscopic environment and variability are directly and biologically connected.

Second, **F** strongly correlates with **M**, while its correlation with **G** and **E** is positive but modest. Previous studies[39,40] suggested that development can serve as a mediator that converts different sources of input to phenotypic variation. If phenotypic variations caused by different sources are shaped through the same developmental system, then the pattern of phenotypic variation should be similar regardless of whether the variation is genetically or environmentally encoded[48]. The correlation between **M** and **F** supports this idea and indicates that the propensity of this developmental system to generate variation shapes **G** and **E**. As circumstantial evidence of this, the correlation between **G** and **E** ($r = 0.66$) roughly corresponds to the product of the correlation between **F** and **G** ($r = 0.81$) and **F** and **E** ($r = 0.79$). This would be expected if **F** causes **G** and **E** independently but in a similar manner.

Third, the pattern of microevolution ($\mathbf{D}_{pop}$) is correlated with **G** with a scaling exponent of 0.80 ($\pm 0.16$ s.e.), which is the closest to 1 among all pairs we examined. The evolutionary genetics theory proposes that the directions and rates of microevolution should be determined by **G**[49–53] and that the exponent between $\mathbf{D}_{pop}$ and **G** should be 1 if drift is the dominant evolutionary force. Conversely, as the influence of local adaptations or long-term fluctuations in adaptive peaks strengthens, the scaling exponent of the variation–divergence relationship should increasingly deviate from 1[8,11]. Moreover, the relationships are globally consistent with the direction of causation from $\mathbf{M} \rightarrow \mathbf{G}$ and $\mathbf{G} \rightarrow \mathbf{D}_{pop}$ because $R^2$ progressively decays in this order as would be expected if the causal hypothesis is correct and measurement errors are added at each step of inference. Therefore, our results are consistent with the established theory of evolutionary genetics and support the idea that genetic constraints dominate the pattern of microevolution.

The following scenarios are proposed to explain our results: (i) macroevolutionary history (**R** and $\mathbf{D}_{sp}$) shapes the developmental system and determines variability (**F** and **M**) through the congruence mechanism; (ii) the developmental system shapes the standing variation (**G** and **E**) under the influence of local adaptation and drift; (iii) **G** determines the pattern of

microevolution ($\mathbf{D}_{pop}$). Based on this hypothesis, macroevolution is often predictable[8,9,11,18] because the long-term pattern of fluctuation in adaptive peaks molds variability. Our hypothesis also explains why microevolution is not always predictable from **G** and contemporary selection[54,55] because neither microevolution, **G**, nor selection are direct descriptors of variability and long-term peak movements.

Allometry is a potential mechanism underlying the strong alignment among co/variance matrices reported in the present study. *Drosophila* wing morphology exhibits allometric variation (i.e., shape variation that covaries with overall size), in terms of the distance between landmarks 4 and 5[56]. It is conceivable that allometry—the growth rate of wing parts relative to the whole wing—converts different sources of perturbations to similar phenotypic outcomes. Recently, Rohner and Berger[57] demonstrated that allometry accounts for about 20% of **F** in a Dipteran species, *Sepsis punctum*. However, they also showed that the strong alignment between **F** and other variance matrices remains after allometric variations are removed from the matrices. Therefore, even though allometry constitutes a major part of wing shape variation in Diptera, the alignment among co/variance matrices across six levels of organization reported here is unlikely to be solely attributable to allometry. Nonetheless, studying how allometry, and the variability of allometric slope in particular[56], influences the relationship between **F** and other co/variance matrices is clearly a fruitful avenue for future research.

## Conclusion

Our data support the congruence mechanism to explain the correlation between macroevolution and the standing variation within a population. Rather than seeking a single explanation, our perspective advocates for a pluralistic view that acknowledges the multifaceted link between microevolution and macroevolution, encompassing multiple evolutionary processes and mechanisms operating at various levels and directions. Although our hypothesis that macroevolution molds variability may appear radical, it is consistent with classic theories of evolutionary genetics[5,6,14,27] and evidence supporting our view is found in the literature of non-genetic inheritance[21–24], systems biology[58], and models of macroevolution[46,47]. Our findings contribute to the development of a unified theory of evolution applicable to all time scales.

## Methods

### Sampling and establishment of isofemale lines

We sampled *Drosophila simulans* from multiple wild locations in Japan, each separated by more than 20 km (Supplementary Fig. 2). Sampling was

conducted through sweeping or by collecting fruits that were likely to have eggs laid on them. Isofemale lines were then established from the sampled adult females (Supplementary Table 2). Each isofemale line was repeatedly inbred over several generations to reduce genetic variation within an isofemale line. During this process, each isofemale line was maintained under identical conditions to eliminate environmental and maternal effects. The condition consists of media described in Fitzpatrick et al.[59] (500 mL of $H_2O$, 50 g of sucrose, 50 g of dry yeast, 6.5 g of agar, 5.36 g of $KNaC_4H_6 \cdot 4H_2O$, 0.5 g of $KH_2PO_4$, 0.25 g of NaCl, 0.25 g of $MgCl_2$, 0.25 g of $CaCl_2$, and 0.35 g of $Fe_2(SO_4) \cdot 6.9H_2O$) in 170-mL bottles, maintained at 25 °C with a 12 h light :12 h dark cycle. In addition, *D. lutescens* were sampled from a single wild population in Japan (the campus of Chiba University: 35° 62′ 79″ N, 140° 10′ 31″ E), and isofemale lines were established in the same manner as for *D. simulans*.

## Rearing experiments

In quantifying the phenotypic plasticity, previously published data were used[42]. Briefly, larvae of individual *D. simulans* were reared from the egg stage through hatching and metamorphosis to adults under seven combinations of three environmental factors. These combinations consisted of three nutrient conditions (high, intermediate, or low), three light–dark cycle conditions (10L:14D, 12L:12D, or 14L:10D), and three temperature conditions (20 °C, 23 °C, or 26 °C). See Saito et al.[42] for further details of this experiment. In addition, we utilized isofemale lines of *D. simulans* collected from the Chiba University campus and maintained under standard conditions (high, 12L12D, and 25 °C). Consequently, the total number of environmental conditions was eight (Supplementary Table 3).

## Wing collection and datasets

After establishing the isofemale lines, adult females were collected from isofemale lines of *D. simulans* and *D. lutescens*. The left and right wings were separated from their bodies and then directly placed on the glass slide. To flatten the wings, a cover glass was placed on the wings and glued to the glass slide (i.e., dry mount). The wings were imaged using the CMOS camera (Leica MC190 HD, 10 million pixels) of the stereoscopic fluorescence microscope (Leica M165 FC) under the condition in which the wings were illuminated by the Torres stand under the glass slide. In addition, wing photos were collected from previous studies. First, as mentioned previously, the datasets of Saito et al.[42] were used to estimate **E**. Second, wing photos of 13 *Drosophila* species were collected from DrosoWing Project[41] to estimate species divergence. These three wing photo datasets were combined and used for the following analysis. The total number of wing photos was 5569 (Supplementary Table 4), and all wings were obtained only from females. In estimating the narrow range of species divergence matrix, wing photos of 14 *Drosophila* species were used. By contrast, for population divergence, genetic variation, variation caused by phenotypic plasticity, and developmental noise, only *D. simulans* wing photos were used.

## Wing measurements and analyses of wing shapes

In the present study, the *x* and *y* coordinates of 12 landmark vein intersections were measured following Houle et al.[8] from collected photos (Supplementary Fig. 3), and a semi-automated procedure was used to acquire these coordinates. First, the machine-learning program, "*ml-morph*"[60], was used to obtain the *x* and *y* coordinates of landmarks automatically. Next, considering that the *x* and *y* coordinates of 12 landmarks extracted from "*ml-morph*" were rarely incorrect, these *x* and *y* coordinates were manually corrected by using "*imglab*" of "*Dlib*." We also used the training set for *ml-morph* described in Saito et al.[42]. This procedure was performed twice to evaluate the effect of measurement error. Finally, the *x* and *y* coordinates were geometrically aligned to eliminate the variation caused by wing size, wing direction, and the magnification of the camera when shooting using Generalized Procrustes Analysis (GPA). This procedure translates original *x* and *y* coordinate data to a common coordinate system by keeping constant variation in their position, size, and direction.

We performed GPA using the "*geomorph*" package in $R$[61]. These standardized coordinates were used to estimate each co/variance matrix.

## Estimating variance matrices by MCMCglmm

A Bayesian mixed-effect model implemented in the MCMCglmm package version 2.35[62] was used to estimate the co/variance matrices at five levels: $D_{sp}$, $D_{pop}$, **G**, **E**, and **F**. In all models except for the model estimating **F**, standarized *x* and *y* coordinates averaged across four measurements in each individual (i.e., repeated measurements of left and right wing) were analyzed as the multivariate response variable of the model, and a flat uninformative prior was used. Four models were run, each with the random effects of species, population, treatment, or ID of isofemale lines to estimate $D_{sp}$, $D_{pop}$, **E**, and **G**, respectively. In a model estimating **F**, all measurements (i.e., four measurements per individual) were used as the multivariate response variable in which individual and a dummy variable that indexes individual ID and side (left or right) were modeled as the random effect. We did not include side as the fixed effect to estimate directional asymmetry in this model because preliminary analysis showed no effect of side (Supplementary Table 5). The resultant variance component associated with this dummy variable of the model was used to evaluate FA accounting for instrumental measurement error. As a prior of this model, a matrix whose diagonals are the empirically estimated **F** reported in Saito et al.[42] and off-diagonals are zeros was used. All models were run for 750,000 iterations with burn-in and thinning intervals of 250,000 and 5000, respectively, to yield 1000 posterior samples. Model convergence was assessed by evaluating the posterior, and chain mixing was tested formally in accordance with the Heidelberg criteria[63] where all models passed the test. The details of our analysis are described in lines 21 to 198 of our R code.

## Estimated matrices

For a wider range of species divergence, the rate of macroevolution matrix (**R**) was used. This matrix was estimated in Houle et al.[8]. The species divergence we estimated with data from online image repository ($D_{sp}$) represents the variance in trait means among species, without considering their phylogenetic relatedness. In contrast, **R** (taken from Houle et al.[8]) is estimated from a phylogenetic mixed model. Assuming a multivariate Brownian motion, this model estimates **R** as the component of variance in trait means among species that are attributable to phylogenetic relatedness. Hence, the two matrices describing species divergence (wider taxonomic range using **R** vs. narrow taxonomic range using $D_{sp}$) differ in whether the matrices considered phylogenetic information. **R** is often considered a more formal estimate of the rate of macroevolution than the raw divergence ($D_{sp}$). Conceptually, the use of $D_{sp}$ was included in our analyses to test the established idea (e.g., Holstad et al.[11], Tsuboi et al.[12], Rohner and Berger[57]) that the correlation between variation and divergence should decay progressively as wider taxa are considered, if genetic constraints are the cause of this relationship. In addition, **M** ($M_{hom}$ and $M_{het}$) in *D. melanogaster* was used and estimated in accordance with the methods of Houle and Fierst[38] and Houle et al.[8] to investigate the subset phenotypic variance attributable to spontaneous mutation.

## Comparison

All co/variance matrices except for **R** and **M** were the median of the posterior distribution of co/variance matrices obtained by corresponding models. To compare co/variance matrices, each matrix was projected to the eigenvector of the additive genetic co/variance matrix (**G**) of *D. melanogaster* presented in Houle et al[8]. Each projected matrix has 24 traits (12 landmarks of *x* and *y* coordinates), but performing GPA reduces the number of dimensions by 4. Therefore, the trait-sets corresponding to the first 20 eigenvectors of *D. melanogaster* **G** were used. Since $M_{hom}$ and $M_{het}$ were less than the full rank[8], only upper 17 dimensions of $M_{hom}$ or eight dimensions of $M_{het}$ were used (see Houle et al[8]. for the rationale). In examining the relationship among each matrix, the ordinary least-squares regression was used, and the coefficient of determination ($R^2$) and the slope

score ($\beta$) were calculated. Each analysis was based on Houle et al.[8] and Rohner and Berger[18].

## Statistics and reproducibility

All statistical analyses were performed using *R* 4.1.2, with detailed descriptions provided in the Methods section. Each *Drosophila* wing specimen was photographed twice to ensure measurement reliability. The map figure was generated using QGIS, an open-source geographic information system software.

## Reporting summary

Further information on research design is available in the Nature Portfolio Reporting Summary linked to this article.

## Data availability

All raw data generated and analyzed during this study are available on Figshare under the following https://doi.org/10.6084/m9.figshare.26861356[64]. The source data underlying all main figures are provided as Supplementary Data. All other relevant data are available from the corresponding author upon reasonable request.

## Code availability

All code used in this study is available on Figshare under the following DOI: 10.6084/m9.figshare.26861356[64]. The parameters used for the analyses are included in the code, and the software versions are described in the Methods section.

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

## Acknowledgements

We thank Tobias Uller, Lisandro Milocco, Kunihiko Kaneko, and Nobuto Takeuchi for their insightful comments on our earlier versions of this work. We also thank four anonymous reviewers whose constructive criticisms have improved the clarity of our study. The work was supported by a part of KAKENHI (grant no. 20H04857) awarded to Y.T.

## Author contributions

All authors. Methodology: All authors. Investigation: K.S. Visualization: K.S. and M.T. Funding acquisition: Y.T. Project administration: All authors. Supervision: M.T. and Y.T. Writing – original draft: K.S. Writing – review & editing: All authors.

## Funding

## Competing interests

The authors declare no competing interests.
