## [Transparent Peer Review file · Communications Biology]

Conserved wing shape variation across biological scales unveils dialectical relationships between micro- and macroevolution

Corresponding Author: Dr Yuma Takahashi

This manuscript has been previously submitted at another journal. This document only contains information relating to versions considered at Communications Biology.

Version 0:

Reviewer comments:

Reviewer #1

(Remarks to the Author)

Saito et al set out to model the relationships between micro and macro evolution using wing shape variation among species of *Drosophila*. They found that micro and macro- evo variation are positively correlated over long time spans. But somewhat surprisingly, they also found that the strongest correlation was between “developmental noise” and “macroevolutionary divergence”. Conversely they observed that “standing variation” was not as strongly correlated with “population divergence”, which is commonly predicted to be the case.

They build the case for contrasting models of the “constraint hypothesis”, where standing variation predicts divergence, and the “congruence hypothesis”, where variation mirrors shifts in adaptive peaks over long time scales. Under their model, if the constraint hypothesis holds then genetic variance and population divergence should have high correlation, and if the congruence hypothesis holds, then R (wider species divergence) and F (as measured by asymmetry) should be highly correlated.

The authors ultimately conclude that while G and Dpop are indeed correlated, that R and F show the highest correlation in their dataset, lending support for the congruence hypothesis in this scenario, stating that “standing variation is a consequence, rather than a cause, of macroevolution”. They contextualise this in their conclusion by advocating for a pluralistic view, presumably with roles for both constraint and congruence. They also are very clear that they are using their interpretation of congruence to explain apparently non-genetic variation with genetics, within the scope of the modern synthesis.

I found the paper to be well constructed and relatively clear, and while reading the body of the text I held some concerns about overstatement, I think the discussion and concluding remarks appropriately couch their interpretation of their results as a piece of a broader framework rather than as a radical replacement for well established theory. As such, I think it is well placed to contribute to theoretical discussions around evolutionary processes and is appropriate for publication. I have one major comment on interpretation of the variable F, which I share below and which may be addressable through clarifying edits in the text.

As I understand it, F is measured as Fluctuating Asymmetry, i.e. random differences between size / morphology of the left and right side of an individual. In lines 112-116, the authors summarize their use of F thus: “Given that genotype and environmental parameters are fixed when measuring M and F, M captures the effect of de novo mutation on phenotype during the developmental process, whereas F captures the effect of intrinsic developmental perturbation on phenotype during the developmental process”. The idea that genotype is fixed during these measurements seems to be an impossibility to me unless individuals are clones; even very minor genetic variation and subsequent interactions with environment will influence F and confound it with G. What I really want to be clearer in this manuscript is a) what are the measures taken to avoid such a confounding factor and b) if it were indeed a confound, what influence would this have on the outcome and interpretation of the results. (In fact, all of E, F, G and M could be subject to this issue).

I could speculate on whether some of my critique above could come from an ignorance of standard practices in *Drosophila* evolutionary genetics (as I am not a part of that field myself), though the authors may like to note that addressing these comments textually could greatly assist a broader audience to engage with this work.

Reviewer #2

(Remarks to the Author)

Conserved wing shape variation across biological scales unveils dialectical relationships between micro- and macroevolution

This manuscript is asking a fundamental question in evolutionary quantitative genetics, namely, what is the relationship between the different sources of phenotypic variation and evolution at various timescales? To do this, the authors are testing the relationships between different matrices of phenotypic covariance resulting either from mutations, developmental instability, standing genetic variation or environmental variation and matrices of population and species divergence. All matrices are positively correlated but most surprisingly, the strongest correlation is observed between the variation generated by developmental instability and species divergence. The authors interpret these results as evidence for the developmental process to be molded by long term variation in selective optima that generates macroevolution. This variation in the developmental processes in turn mold patterns of genetic and environmental variation.

This manuscript based on high quality data provides an interesting and novel perspective to the link between micro and macroevolution. It is therefore of high interest for evolutionary biologist. I have, however, several comments about this manuscript.

General comments

I find interesting the idea to distinguish between competing hypotheses (constraint and congruence) using the scaling relationship expected between the different measures of variability and the realized evolution (populations or species divergence). However, I think the author underuse this approach. Indeed, in lines 83-95 they develop some predictions about these scaling relationships, but they never fully develop this aspect and they do not clearly state how scaling relationships should differ between the two hypotheses. Making these predictions clearer would greatly improve the ms (see for example Bolstad et al. 2014). This issue is approached again lines 128 and 130, but I am not certain that the argument there is correct since the scaling exponent will depend on the evolutionary process generating the divergence among population or species.

On the same topic, I also found the explanation of the new theory (Lines 88 – 120) unclear. Unfortunately, I did not have access to the paper by Kaneko & Furusawa (2018), and I could only read the abstract. From this, I wonder the relevance of this model to explain the link between the different variance matrices. First I don't understand what is the evolutionary force driving evolution in this model (the equivalent of drift in the model by Lynch and Hill). This is essential since the scaling between variability and realized evolution depends on such process (see Bolstad et al 2014). Additionally if the proportionality in the L&H model is clearly indicated (line 87), it is not explained what proportionality is expected under the K&F model and why such proportionality should occur. One can therefore wonder what should be the scaling exponent between F and R? (see line 132). This model needs to be explained more clearly, but I also notice that the equation underlying the model (line 92) seems incorrect because it states that X is proportional to X². This makes me rather suspicious of the general validity of the argument.

Overall, I wonder whether the reference to the K&F 2018 model is necessary to explain the expected relationship between developmental variance measured as FA and genetic variation and divergence. I feel that this part could be removed without loss.

Some kind of table should be included in the main text to explain clearly where the different matrices are coming from, which species are included in the different divergence matrices, which species is represented in the M-matrix, how many individuals are included in the different matrices (among species, among population and within population). I did not manage to find a clear sentence explaining on which species G and E have been estimated. It seems that this is on simulans (lines 175 178), but it remains unclear. Currently the info is scattered across subsections and in the supplementary material. A table in the main text would greatly help.

Concerning the origin of the different matrices, it seems potentially problematic to start the whole analysis by using M in *melanogaster* and F in *simulans* to test their relationship. Is it not possible to find FA data on *melanogaster*? (several papers by Houle's group seems to have such data). This would be more elegant. If this is not possible, it may be important to explain whether or not this may affect the results.

Additionally, the different analyses/experiments performed in this study to obtain the matrices are not always clearly explained. For several matrix estimations, the authors only refer to other papers without explaining the analyses (e.g. Lines 304 – 306, Line 320). This is problematic and a summary of the methods used should be presented. I also find surprising that P, the phenotypic variance matrix, is never included in the analysis.

Please explain clearly the difference between broad and narrow range of species divergence also in the method section and why such distinction is necessary for the distinction between the two hypotheses. Although I understand that the format of the published paper obliged some info about the method to be given in the result section, this should not prevent the authors to give a complete explanation in the methods.

My main concern or question regarding the methods is the way the M (mutational) matrix is used in the study. The authors

are first testing the relationship between the M-matrix and the F (developmental instability) matrix. Then considering that both matrices are correlated, they only use F to compare to G, D and R. Why not also including the comparison between M and G, D and R? I agree such comparison has been previously done in Houle et al. 2017, but since the data presented here are different for a large part, it could be particularly interesting to repeat the result from this seminal study. Therefore, I think that the comparison of M with the other divergence and genetic matrices should be presented.

Removing size variation from the data – Lines 275 – 276 it is explained that size was removed from the data. This may be problematic if there is an allometric relationship (i.e. not isometric) between wing shape and wing size and if the different species greatly differ in wing size. This choice of method should be better justified. Additionally, directional asymmetry for wing size has been observed in *Drosophila* species before. Although this may not affect the results, it may still be better to show that variation in size does not influence variation in shape and that the possible presence of size DA does not influence the pattern of within individual variation in size.

Minor comments

I am not sure what the authors are meaning with the word “dialectical”. After checking the various meanings of dialectical, I still don’t understand. It may be better to use a simpler and more descriptive title.

Lines 184 – 190: I don’t understand the argumentation and its logic.

Line 233 – 234: The argument is not logical. It is not by decreasing the level of genetic variation via inbreeding that one is removing the environmental variation and maternal effect. These effects are mostly decreased by maintaining individuals in constant environment. Please adjust the text to avoid confusion.

Lines 245 – 249: 3 factors with 3 levels each should give 27 combinations, so what are the 7 combination selected and why those?

Line 274: Replace “to” by “two” (two times). Additionally repeating the measurement does not “prevent” measurement error, but it allows estimating measurement error. Please correct and explain this better in the part concerning the F-matrix.

Reviewer #3

(Remarks to the Author)

I read this manuscript with interest – the topic is of broad interest, and the results are likely to generate noise in the relevant communities. As someone who is largely interested in the macro- side of things with a background in comparative phylogenetic analysis, I cannot comment on the approaches taken to measure genetic variation and plasticity - although at face value, they seem reasonable to me. In general, I feel that this manuscript is largely suitable for publication in *Communications Biology*, but would recommend a few minor revisions in order to improve the quality. Mostly, these are revisions that involve adding detail for the uninformed reader. Please see my detailed comments below.

The introduction of the present manuscript is very well written but clearly targeted to a journal with very short format – I think that it could benefit from a bit of expansion particularly with regard to the ‘theoretical frameworks of quantitative genetics and statistical physics as our guiding principles’ (lines 67-68). At the end of the introduction, I am left to understand that the authors will be looking at different levels of trait variation expressed as VCVs but not really understanding why. With some expansion on what these principles are and why they are relevant to what the authors are studying, this could become much clearer. The authors leave much of this in the beginning of the results section, but I think this could be summarized succinctly up front. Particularly, I would have appreciated the authors laying out explicit expectations. The authors suggest that their major aim is “to elucidate the mechanisms underlying the relationships among variability, variation, microevolution, and macroevolution”. This sounds exploratory and vague. What exactly would the authors expectations be based on the background they have presented in the introduction? What relationships would they expect to find – why? Why would some relationships be unexpected? Perhaps a figure depicting the possible associations and their interpretations could be useful here. At the absolute minimum, the expectations for the relationships between each parameter given the constraint and congruence hypotheses needs to be outlined clearly.

It is not a new idea that macroevolutionary phenomena can be thought of as the average effect of patterns observed along individual lineages – as mentioned by the authors. This has been demonstrated in various ways – as discussed in the introduction of the present manuscript. However, there’s no current mention about interpretation of things like macroevolutionary regression parameters. I think it would be nice to include a brief discussion / mention of papers that explicitly talk about this in the context of phylogenetic evolution e.g. Hansen et al (2012).

Given the importance of the scale of variances being measured, I would expect more information about the divergences and distances amongst the sampled populations. There is very limited information about which populations were sampled – and even less so about why or how. This doesn’t need to be extensive, but just a bit of contextual information would be helpful.

Hansen, Thomas F., and Krzysztof Bartoszek. "Interpreting the evolutionary regression: the interplay between observational and biological errors in phylogenetic comparative studies." *Systematic biology* 61.3 (2012): 413-425

Minor comments:

L121: constraint or constraints, not constrains

L269: Figure?

L283: No citation for MCMCglmm

L282-300 Refer reader to R code – but these models are beautifully described!

L299-300: Did replicate analyses converge on the same answers?

Version 1:

Reviewer comments:

Reviewer #1

(Remarks to the Author)

I'm satisfied that the authors have appropriately revised their manuscript.

Reviewer #3

(Remarks to the Author)

As before, I read the manuscript with much interest. I feel that with the authors revisions, they have substantially improved not only the quality but also the readability. They have addressed all of my questions in full. Therefore, I am happy to say that this manuscript is suitable for publication in *Communications Biology*.

Reviewer #4

(Remarks to the Author)

Review of the Revised Manuscript entitled "Conserved wing shape variation across biological scales unveils dialectical relationships between micro- and macroevolution", by Saito et al.

I think that the authors have thoroughly and thoughtfully addressed the reviewers' comments in this revised version of the manuscript. I have carefully examined both, the revised manuscript and the detailed rebuttal letter, and I find that the authors have responded appropriately and effectively to the concerns raised during the first round of review.

In particular:

- Respect to the interpretation of fluctuating asymmetry (F), I believe that the authors acknowledge the potential confounding effects of minor genetic and environmental variation and clarify their assumptions and methods. They support their position with references to relevant empirical and theoretical studies and incorporate these clarifications directly into the text. Their response seems balanced, transparent, and in line with current standards in evolutionary quantitative genetics.

- Regarding the Reviewer 2's more extensive critique, I think that the authors significantly improved the clarity of the conceptual framework, especially regarding scaling exponents and the contrasting predictions of the hypotheses they tested (i.e., constraint and congruence hypotheses). They expanded their discussion of the theoretical basis (including Kaneko & Furusawa's model), clarified the origins and estimation methods for all covariance matrices, and included a new summary table that greatly improves readability. Also, methodological details were added or clarified throughout the manuscript and supplemental material. Finally, it seems that the authors addressed the concerns about allometry and directional asymmetry in a proper manner.

Overall, I may say that the authors have strengthened both the scientific rigor and the accessibility of the manuscript. I think that the changes made through the revision process enhanced the manuscript's value which seems a significant contribution to theoretical discussions on the relationship between micro- and macroevolutionary processes.

Minor issues

I recommend reading through the manuscript carefully to correct minor grammatical issues throughout the text. Below, I mention some specific examples to illustrate the types of corrections needed.

- Lines 83-84: "First, based on the quantitative genetic model of Lande proposed the following relationship". I believe this sentence should be rewritten, for example, as follows: "First, based on the quantitative genetic model of Lande, we propose the following relationship".

- Line 165: "while such causal relationship should become progressively decay". Again, I think this sentence should be rewritten, for example, as follows: "should progressively decay" or "should become progressively weaker".

- Line 337: "and all wings were obtained only from female" should be "...from females".

- Lines 387-388: "a more formal estimates of the rate of macroevolution than the raw divergence" should probably be "a more formal estimate of the rate..."

- Table 1: "Drosophila Simulans" should be *D. simulans*.

- Lines 613-614: "photo" should be "photos".

- Line 617: "It includes both sides of wing and replications" should be rewritten, for example, as follows: "It includes both wings and replicated measurements".

Lines 132-136: At this point in the manuscript, it may be helpful to briefly clarify two things:

- Why M was estimated using *D. melanogaster* while F was estimated using *D. simulans*?

- Why two estimates of M were used (i.e., M_{hom} and M_{het})?

Even a concise explanation (acknowledging that more details may follow in the Methods or Supplementary Materials) might help readers to better understand the rationale for these choices, which are central to the comparisons being made.

Lines 155-158: As with the earlier mention of M and F being estimated from different species almost without explanation, it would be helpful to briefly clarify here what is meant by “our dataset” and, perhaps, how it relates to the other data sources mentioned (e.g., *D. simulans* populations, the DrosoWing Project, and Houle et al.'s dataset). This would help readers understand the structure and origin of the data used for estimating divergence at different evolutionary scales.

Dear Editors and Reviewers

We appreciate the opportunity to submit our revised manuscript titled “Conserved wing shape variation across biological scales unveils dialectical relationships between micro- and macroevolution”. We are sincerely thankful for the valuable suggestions provided by the reviewers. Many of their comments highlighted the need for clarification in various sections of our manuscripts, particularly the introduction and methods. We have diligently incorporated almost all these suggestions, significantly enhancing our works.

The reviewers’ comments are presented in bold black, while our responses are provided in plain blue text. When we have made modifications to the text in response to the reviewers’ valuable suggestions, we include both the original submission and the revised version for easy comparison.

Responses to the Comments by the Reviews:

Reviewer: 1

Saito et al set out to model the relationships between micro and macro evolution using wing shape variation among species of *Drosophila*. They found that micro and macro- evo variation are positively correlated over long time spans. But somewhat surprisingly, they also found that the strongest correlation was between “developmental noise” and “macroevolutionary divergence”. Conversely they observed that “standing variation” was not as strongly correlated with “population divergence”, which is commonly predicted to be the case.

They build the case for contrasting models of the “constraint hypothesis”, where standing variation predicts divergence, and the “congruence hypothesis”, where variation mirrors shifts in adaptive peaks over long time scales. Under their model, if the constraint hypothesis holds then genetic variance and population divergence should have high correlation, and if the congruence hypothesis holds, then R (wider species divergence) and F (as measured by asymmetry) should be highly correlated.

The authors ultimately conclude that while G and Dpop are indeed correlated, that R and F show the highest correlation in their dataset, lending support for the congruence hypothesis in this scenario, stating that “standing variation is a consequence, rather than a cause, of macroevolution”. They contextualise this in their conclusion by advocating for a pluralistic view, presumably with roles for both constraint and congruence. They also are very clear that they are using their interpretation of congruence to explain apparently non-genetic variation with

genetics, within the scope of the modern synthesis.

I found the paper to be well constructed and relatively clear, and while reading the body of the text I held some concerns about overstatement, I think the discussion and concluding remarks appropriately couch their interpretation of their results as a piece of a broader framework rather than as a radical replacement for well established theory. As such, I think it is well placed to contribute to theoretical discussions around evolutionary processes and is appropriate for publication. I have one major comment on interpretation of the variable F, which I share below and which may be addressable through clarifying edits in the text.

We thank the Reviewer for the positive and constructive comments and suggestions. Our detailed responses are below.

As I understand it, F is measured as Fluctuating Asymmetry, i.e. random differences between size / morphology of the left and right side of an individual. In lines 112-116, the authors summarize their use of F thus: “Given that genotype and environmental parameters are fixed when measuring M and F, M captures the effect of de novo mutation on phenotype during the developmental process, whereas F captures the effect of intrinsic developmental perturbation on phenotype during the developmental process”. The idea that genotype is fixed during these measurements seems to be an impossibility to me unless individuals are clones; even very minor genetic variation and subsequent interactions with environment will influence F and confound it with G. What I really want to be clearer in this manuscript is a) what are the measures taken to avoid such a confounding factor and b) if it were indeed a confound, what influence would this have on the outcome and interpretation of the results. (In fact, all of E, F, G and M could be subject to this issue).

Thank you so much for your feedback. As you pointed out, the developmental noise we measured contains minor genetic variation and slight differences in rearing environments. Therefore, it is possible that our developmental noise (F) may have been overestimated compared to the true developmental noise. However, these effects are uncontrollable, and we cannot exclude them from our experimental system.

On the other hand, based on several pieces of evidence, we concluded that our F represents biologically significant developmental noise. Specifically:

- a) Our results are consistent with studies that use fluctuation asymmetry to estimate F (e.g., Rohner and Berger, 2024).
- b) We observed similar trend to theoretical studies that have measured accurate developmental noise (e.g., Kaneko and Furusawa, 2018)

c) The heritability of developmental noise is low (e.g., Saito et al., 2024).

I could speculate on whether some of my critique above could come from an ignorance of standard practices in *Drosophila* evolutionary genetics (as I am not a part of that field myself), though the authors may like to note that addressing these comments textually could greatly assist a broader audience to engage with this work.

We will respond sincerely to the comments provided and strive to make this research more understandable to a broader audience. Thank you for your valuable feedback.

Reviewer: 2

This manuscript is asking a fundamental question in evolutionary quantitative genetics, namely, what is the relationship between the different sources of phenotypic variation and evolution at various timescales? To do this, the authors are testing the relationships between different matrices of phenotypic covariance resulting either from mutations, developmental instability, standing genetic variation or environmental variation and matrices of population and species divergence. All matrices are positively correlated but most surprisingly, the strongest correlation is observed between the variation generated by developmental instability and species divergence. The authors interpret these results as evidence for the developmental process to be molded by long term variation in selective optima that generates macroevolution. This variation in the developmental processes in turn mold patterns of genetic and environmental variation.

This manuscript based on high quality data provides an interesting and novel perspective to the link between micro and macroevolution. It is therefore of high interest for evolutionary biologist. I have, however, several comments about this manuscript.

We appreciate the Reviewer for your positive opinion and important suggestion. Below are our detailed responses.

General comments

I find interesting the idea to distinguish between competing hypotheses (constraint and congruence) using the scaling relationship expected between the different measures of variability and the realized evolution (populations or species divergence). However, I think the author underuse this approach. Indeed, in lines 83-95 they develop some predictions about these scaling relationships, but they never fully develop this aspect and they do not clearly state how scaling relationships should differ between the two hypotheses. Making these predictions

clearer would greatly improve the ms (see for example Bolstad et al. 2014). This issue is approached again lines 128 and 130, but I am not certain that the argument there is correct since the scaling exponent will depend on the evolutionary process generating the divergence among population or species.

Thank you very much for this insightful comment. The reviewer is spot-on regarding the predictions based on the scaling exponent on the variance-divergence relationship presented in Bolstad et al., (2014), which were later further extended in Houle et al., (2017), Voje et al., (2022) and Holstad et al., (2024). We agree with the reviewer that our original formulation regarding our prediction was incomplete and needs further clarifications. Our idea is based on two lines of thoughts. The first is based on the exponent, which provides a series of predictions according to a series of evolutionary processes (drift, linear selection, stabilizing selection). The second line is based on the coefficient of determination (R^2) which is affected by (1) measurement error and (2) biological causes. To clarify these points, we have extended the explanation for our predictions, which are found in line 81-100 and line 158-167 of our revision.

On the same topic, I also found the explanation of the new theory (Lines 88 – 120) unclear. Unfortunately, I did not have access to the paper by Kaneko & Furusawa (2018), and I could only read the abstract. From this, I wonder the relevance of this model to explain the link between the different variance matrices. First I don't understand what is the evolutionary force driving evolution in this model (the equivalent of drift in the model by Lynch and Hill). This is essential since the scaling between variability and realized evolution depends on such process (see Bolstad et al 2014). Additionally if the proportionality in the L&H model is clearly indicated (line 87), it is not explained what proportionality is expected under the K&F model and why such proportionality should occur. One can therefore wonder what should be the scaling exponent between F and R? (see line 132). This model needs to be explained more clearly, but I also notice that the equation underlying the model (line 92) seems incorrect because it states that X is proportional to X². This makes me rather suspicious of the general validity of the argument.

Overall, I wonder whether the reference to the K&F 2018 model is necessary to explain the expected relationship between developmental variance measured as FA and genetic variation and divergence. I feel that this part could be removed without loss.

Thank you for your insightful comments. We reviewed the paper by Kaneko and Furusawa (2018) and add more explanation about the theory.

Original Text (Line 93-94): where the variance of the phenotypic trait X for a given

system is parameterized by a , which is assigned as a parameter that specifies the genotype in this study.

New Text (Line 113-121): where $\langle X \rangle_a$ and $\langle (\delta X)^2 \rangle = \langle (X - \langle X \rangle)^2 \rangle_a$ represent the average and variability, respectively, of the phenotypic trait X for a given system parameterized by a . The relationship described above was derived under the assumption that the distribution $P(X; a)$ follows an approximately Gaussian form, with the effect of changes in a on the distribution represented by a bilinear coupling between X and a . When a is assigned as a parameter that specifies the genotype, the left-hand side of the equation, $(\langle X \rangle_{a+\Delta a} - \langle X \rangle_a) / \Delta a$, quantifies the phenotypic change resulting from a genetic alteration and represents the rate of evolution (\mathbf{R})³³. Therefore, the above relationship can be reinterpreted as a proportional relationship between phenotypic fluctuations due to developmental noise and the rate of evolution.

Some kind of table should be included in the main text to explain clearly where the different matrices are coming from, which species are included in the different divergence matrices, which species is represented in the M-matrix, how many individuals are included in the different matrices (among species, among population and within population). I did not manage to find a clear sentence explaining on which species G and E have been estimated. It seems that this is on simulans (lines 175 178), but it remains unclear. Currently the info is scattered across subsections and in the supplementary material. A table in the main text would greatly help.

Concerning the origin of the different matrices, it seems potentially problematic to start the whole analysis by using M in melanogaster and F in simulans to test their relationship. Is it not possible to find FA data on melanogaster? (several papers by Houle's group seems to have such data). This would be more elegant. If this is not possible, it may be important to explain whether or not this may affect the results. Additionally, the different analyses/experiments performed in this study to obtain the matrices are not always clearly explained. For several matrix estimations, the authors only refer to other papers without explaining the analyses (e.g. Lines 304 – 306, Line 320). This is problematic and a summary of the methods used should be presented. I also find surprising that P, the phenotypic variance matrix, is never included in the analysis.

Thank you for your feedback. We add Table (Line 133, 158, 208).

Table 1. List of used species for estimating matrix and the way of estimating.

Matrix	Used Species	Estimating way	Sample size
R	112 species	From the previous study [‡]	112 species
D_{sp}	14 species*	Our MCMCglmm model	14 species
D_{pop}	Drosophila Simulans	Our MCMCglmm model	11 populations
G	D. simulans	Our MCMCglmm model	33 isofemale lines
E	D. simulans [†]	Our MCMCglmm model	8 environments
F	D. simulans	Our MCMCglmm model	7,654 wings [¶]
M	D. melanogaster	From the previous study [§]	12,075 wings

*It includes the wing photo obtained from DrosoWing Project.

†It includes the wing photo obtained from Saito et al. (2024).

‡Houle et al. (2017).

§Houle and Fierst (2013) and Houle et al. (2017).

¶It includes both sides of wing and replications.

F cannot be estimated from the Houle group's data because they did not measure both sides of the wings. As you pointed out, measuring **M** in *D. simulans* would yield more accurate results. However, estimating **M** requires conducting Mutation Accumulation (MA) experiments (we mentioned it in Line 126), which are time-consuming in *Drosophila* due to their longer generation time compared to organisms such as *Escherichia coli*, which are commonly used for MA experiments. Therefore, in this study, we used **M** from *D. melanogaster*, a species closely related to *D. simulans*. Nevertheless, we acknowledge the importance of comparing **M** with other matrices, and thus, we have included additional results in the Supplementary information.

New Text (Line 171-172): In addition, to avoid the potential impact of using different species' **M**, we conducted additional comparisons between **M** and the other matrices (Supplementary Fig. 1).

Supplementary Fig. 1 Relationships between mutational variations in *Drosophila melanogaster* (M_{hom} and M_{het}) and our estimated matrix (E, G, D_{pop} , and D_{sp}).

Points represent \log_{10} (variance in each matrix) along the eigenvectors of G in *Drosophila melanogaster*. Key gives log-log regression result, $\beta \pm s.e.$ and R^2 . Only upper 17 dimensions of M_{hom} and eight dimensions of M_{het} were used. We did not compare both M with R because these relationships were already revealed in Houle et al. 2018.

In this study, we intentionally exclude **P** from the analysis. Phenotypic variance is composed of genetic variance, plasticity, developmental noise, and error ($P = G + E + F + \text{error}$). Therefore, we argue that including **P** in the analyses would not provide significant additional information for our discussion.

Please explain clearly the difference between broad and narrow range of species divergence also in the method section and why such distinction is necessary for the distinction between the two hypotheses. Although I understand that the format of the published paper obliged some info about the method to be given in the result section, this should not prevent the authors to give a complet explanation in the methods.

Thank you for the comment, we add more information about it.

New Text (Line 380-392): The two matrices describing species divergence (wider taxonomic range using **R** vs. narrow taxonomic range using **D_{sp}**) differ in whether the matrices considered phylogenetic information. The species divergence we estimated with data from online image repository (**D_{sp}**) represents the variance in trait means among species, without considering their phylogenetic relatedness. In contrast, **R** (taken from Houle et al.⁸) is estimated from a phylogenetic mixed model. Assuming a multivariate Brownian motion, this model estimates **R** as the component of variance in trait means among species that are attributable to phylogenetic relatedness. **R** is often considered a more formal estimates of the rate of macroevolution than the raw divergence (**D_{sp}**). Conceptually, the use of **D_{sp}** was included in our analyses to test the established idea (e.g., Holstad et al.¹¹, Tsuboi et al.¹², Rohner and Berger⁵⁷) that the correlation between variation and divergence should decay progressively as wider taxa are considered, if genetic constraints are the cause of this relationship.

My main concern or question regarding the methods is the way the M (mutational) matrix is used in the study. The authors are first testing the relationship between the M-matrix and the F (developmental instability) matrix. Then considering that both matrices are correlated, they only use F to compare to G, D and R. Why not also including the comparison between M and G, D and R? I agree such comparison has been proviously done in Houle et al. 2017, but since the data presented here are different for a large part, it could be particularly interesting to repeat the result from this seminal study. Therefore, I think that the comparison of M with the other divergence and genetic matrices should be presented.

We appreciate this suggestion. As mentioned in my other response to the comment, we conducted additional analyses to compare **M** with other matrices and added the results in the Supplementary information.

Removing size variation from the data – Lines 275 – 276 it is explained that size was removed from the data. This may be problematic if there is an allometric relationship (i.e. not isometric) between wing shape and wing size and if the different species greatly differ in wing size. This choice of method should be better justified. Additionally, directional asymmetry for wing size has been observed in *Drosophila* species before. Although this may not affect the results, it may still be better to show that variation in size does not influence variation in shape and that the possible presence of size DA does not influence the pattern of within individual variation in size.

Thank you for your suggestion. In this study, size variation was removed due to the use of different camera magnifications when photographing the wings for clarity. We could have included size, but due to logistical limitation, it was not feasible for this study.

That said, we do have several lines of evidence suggesting that the exclusion of size from our analyses is unlikely influence our results and overall arguments. First, according to the study on the allometry of *Drosophila* wings by Bolstad et al. (2015), the relationship between wing shape and size is generally isometric (i.e., the slope of log-log regression is 1), although there are a few species whose allometric coefficients deviate from isometry. This suggests that the effect of wing size on wing shape should be small.

Second, recent publication by Rohner and Berger (2025) discussed the effects of allometry on Dipteran wing morphology. They presented (1) that allometry contributes to 20% and 25% of the variance in **F** and **R**, and (2) even when these components are removed from the matrices, the strong alignment between matrices at different levels remains. Thus, in Dipteran wing, the alignment is supported with or without allometry.

Third, in line with the general conclusion of Rohner and Berger (2025), the most favored argument, based on all evidence that we have thus far, appears that allometry does have an effect on the co/variances of wing shape in *Drosophilidae*, but it does so in a similar manner for all levels of biological organization—from **F** to **R**. If this hypothesis is correct, it is unlikely that the effect of allometry differs across biological levels (e.g., the allometry affect **F** but not **G**).

To address these points, we have now extended our discussion (Line 274-286).

Related to directional asymmetry (DA), we mentioned that we considered it in our analyses in the paragraph of MCMCglmm, but we did not show the result. Now, we provide estimates of DA in Supplementary information. In short, the evidence for DA is

essentially non-existing, with the average of signed differences between left and right side of the wing traits (x-y coordinates of 12 landmarks) fall between the range of -4.6% (the right side has greater values than the left side) and 1.8% of the trait value, with the median of 0.20%, meaning that there are no tendencies for left and right side landmarks to be consistently biased.

Supplementary Table 5. The estimate of directional asymmetry.

trait	post.mean	l-95% CI	u-95% CI
traitx1:sideR	0.0003793	0.0001893	0.0005748
traity1:sideR	0.00005429	-0.000237	0.0003266
traitx2:sideR	-4.402E-05	-0.0002717	0.0001972
traity2:sideR	-0.0004754	-0.0006093	-0.0003383
traitx3:sideR	0.00009469	-0.000185	0.0003652
traity3:sideR	0.00028	0.0001203	0.000449
traitx4:sideR	0.0003155	-0.0000436	0.0006064
traity4:sideR	0.0004039	0.0002326	0.0005944
traitx5:sideR	-0.0002957	-0.000466	-0.0001362
traity5:sideR	0.0004851	0.0002727	0.000704
traitx6:sideR	-0.000173	-0.0003447	4.827E-06
traity6:sideR	0.00007076	-0.0001411	0.0002874
traitx7:sideR	-5.875E-05	-0.0003682	0.0002545
traity7:sideR	-0.001138	-0.001441	-0.0007987
traitx8:sideR	-0.0004169	-0.0008274	-8.594E-05
traity8:sideR	0.0006223	0.0004694	0.0007823
traitx9:sideR	0.0001203	-4.507E-05	0.0002759
traity9:sideR	0.0005425	0.0004631	0.0006481
traitx10:sideR	0.00003854	-0.0001542	0.0002359
traity10:sideR	-0.0004269	-0.0005074	-0.0003388
traitx11:sideR	0.0002962	0.0001665	0.0004347
traity11:sideR	0.00007966	-2.297E-05	0.0001982
traitx12:sideR	-0.0002553	-0.0003717	-0.0001704
traity12:sideR	-0.0004988	-0.0006166	-0.0004048
traitx1	-0.1347	-0.135	-0.1344
traity1	-0.2	-0.2003	-0.1998
traitx2	-0.4414	-0.4416	-0.4413
traity2	-0.02596	-0.02612	-0.02582
traitx3	-0.4656	-0.4658	-0.4654
traity3	0.05452	0.05435	0.0547
traitx4	-0.2987	-0.299	-0.2983
traity4	0.1833	0.183	0.1836
traitx5	0.2249	0.2248	0.2251

traity5	0.09948	0.0993	0.09967
traitx6	0.3555	0.3553	0.3556
traity6	0.06687	0.06669	0.06703
traitx7	-0.03186	-0.03216	-0.03157
traity7	-0.1099	-0.1102	-0.1097
traitx8	-0.03882	-0.03915	-0.03851
traity8	-0.03994	-0.04007	-0.03981
traitx9	0.1378	0.1376	0.138
traity9	-0.01183	-0.01189	-0.01175
traitx10	0.1363	0.136	0.1365
traity10	0.01563	0.01555	0.0157
traitx11	0.2616	0.2615	0.2617
traity11	0.01798	0.01789	0.01805
traitx12	0.295	0.2949	0.2952
traity12	-0.05005	-0.05013	-0.04996

Minor comments

I am not sure what the authors are meaning with the word “dialectical”. After checking the various meanings of dialectical, I still don’t understand. It may be better to use a simpler and more descriptive title.

Thank you very much for your suggestion. We carefully considered alternative words and decided against changing the use of “dialectical” in our manuscript. We are empathetic to the confusion that the use of this term may have caused to the reviewer because the word “dialectical” is not common in biology and natural sciences. The way we use this term follows a book by Richard Levins and Richard Lewontin “*Dialectical Biologists*” (1985). According to the authors, a dialectical view holds that parts and whole are interdependent, with no predefined directions of causality across levels of organization. In the context of our manuscript, we used the term to signify that our results favor a perspective against privileging any single level—whether it be gene, development, individual, population, species, or lineages—in our explanations of evolution. We particularly would like to highlight the connotation of the term that it refers to “parts and whole” relationship. This connotation fits the core issue in relating levels of biological organization, which is about the organization of parts (gene, cell etc.) and whole (organ, individual, population etc.). Alternative words we considered are “reciprocal”, “mutual”, “interactive” and “dynamic”, but neither of which have the correct connotation that the problem is about the parts-whole relationship. Now, we also add some sentence related to the word “dialectical” in Line 220-228.

Lines 184 – 190: I don’t understand the argumentation and its logic.

Thank you for your comment. We thought through our logic again, and we concluded to change the logic here, because the argument was not strong as the reviewer has correctly pointed out. Now, we have revised our logic of why we conclude the causal relationship between **F** and **R** to be **R** to **F** and not **F** to **R**, based on (1) the argument that measurement errors and the precision of causal hypotheses interact, and (2) the traditional logic from Schluter (1996) that the predictability of contemporary variation should decay over time because co/variance should themselves evolve (Line 229-242).

Line 233 – 234: The argument is not logical. It is not by decreasing the level of genetic variation via inbreeding that one is removing the environmental variation and maternal effect. These effects are mostly decreased by maintaining individuals in constant environment. Please adjust the text to avoid confusion.

Thank you. Now we corrected these sentences.

Original Text (Line 233-239): Each isofemale line was repeatedly inbred over several generations to reduce genetic variation within an isofemale line and to remove environmental and maternal effects. Each isofemale line was maintained in standard media on the basis of the method of Fitzpatrick et al.(Fitzpatrick et al., 2007) (500 mL of H₂O, 50 g of sucrose, 50 g of dry yeast, 6.5 g of agar, 5.36 g of KNaC₄H₆.4H₂O, 0.5 g of KH₂PO₄, 0.25 g of NaCl, 0.25 g of MgCl₂, 0.25 g of CaCl₂, and 0.35 g of Fe₂(SO₄)·6.9H₂O) in 170-mL bottles under a 12 light:12 dark cycle at 25°C.

New Text (Line 305-311): Each isofemale line was repeatedly inbred over several generations to reduce genetic variation within an isofemale line. During this process, each isofemale line was maintained under the identical condition to eliminate environmental and maternal effects. The identical condition consists of standard media on the basis of the method of Fitzpatrick et al.(Fitzpatrick et al., 2007) (500 mL of H₂O, 50 g of sucrose, 50 g of dry yeast, 6.5 g of agar, 5.36 g of KNaC₄H₆.4H₂O, 0.5 g of KH₂PO₄, 0.25 g of NaCl, 0.25 g of MgCl₂, 0.25 g of CaCl₂, and 0.35 g of Fe₂(SO₄)·6.9H₂O) in 170-mL bottles, maintained at 25°C with 12 light:12 dark cycle.

Lines 245 – 249: 3 factors with 3 levels each should give 27 combinations, so what are the 7 combination selected and why those?

Thank you. We found incorrect information of this part. So, now we corrected it. For each environmental condition, one of the three environmental factors varied from the environment 1 (12 h light/12 h dark, 23°C, intermediate nutrient). Since the environment 1 was shared three times, the total number of environmental conditions

was seven. In addition, we also used *D. simulans* maintained in the standard condition. Therefore, the eight combinations were correct. We showed the table of combinations and added this table to Supplementary information.

Supplementary Table 3. The 8 environment combinations for estimating E.

	Nutrients condition	Light-dark cycle	Temperature
Standard	High	12 h light/12 h dark	25°C
Environment 1	Intermediate	12 h light/12 h dark	23°C
Environment 2	High	12 h light/12 h dark	23°C
Environment 3	Low	12 h light/12 h dark	23°C
Environment 4	Intermediate	10 h light/14 h dark	23°C
Environment 5	Intermediate	14 h light/10 h dark	23°C
Environment 6	Intermediate	12 h light/12 h dark	20°C
Environment 7	Intermediate	12 h light/12 h dark	26°C

Line 274: Replace “to” by “two” (two times). Additionally repeating the measurement does not “prevent” measurement error, but it allows estimating measurement error. Please correct and explain this better in the part concerning the F-matrix.

Thank you. We corrected this sentence (Line 348-350).

Reviewer: 3

I read this manuscript with interest – the topic is of broad interest, and the results are likely to generate noise in the relevant communities. As someone who is largely interested in the macro- side of things with a background in comparative phylogenetic analysis, I cannot comment on the approaches taken to measure genetic variation and plasticity - although at face value, they seem reasonable to me. In general, I feel that this manuscript is largely suitable for publication in Communications Biology, but would recommend a few minor revisions in order to improve the quality. Mostly, these are revisions that involve adding detail for the uninformed reader. Please see my detailed comments below.

We thank the Reviewer for their highly evaluation and kind comments. We provide detailed responses below.

The introduction of the present manuscript is very well written but clearly targeted to a journal with very short format – I think that it could benefit from a bit of expansion particularly with regard to the ‘theoretical frameworks of

quantitative genetics and statistical physics as our guiding principles’ (lines 67-68). At the end of the introduction, I am left to understand that the authors will be looking at different levels of trait variation expressed as VCVs but not really understanding why. With some expansion on what these principles are and why they are relevant to what the authors are studying, this could become much clearer. The authors leave much of this in the beginning of the results section, but I think this could be summarized succinctly up front. Particularly, I would have appreciated the authors laying out explicit expectations. The authors suggest that their major aim is “to elucidate the mechanisms underlying the relationships among variability, variation, microevolution, and macroevolution”. This sounds exploratory and vague. What exactly would the authors expectations be based on the background they have presented in the introduction? What relationships would they expect to find – why? Why would some relationships be unexpected? Perhaps a figure depicting the possible associations and their interpretations could be useful here. At the absolute minimum, the expectations for the relationships between each parameter given the constraint and congruence hypotheses needs to be outlined clearly.

We greatly appreciate this insightful comment. As you mentioned it, our original introduction was quite brief. Therefore, we have now expanded it to include a more detailed explanation of the theoretical frameworks and explicit expectations. Additionally, we considered creating a figure to illustrate both hypotheses for improved clarity. However, conveying these hypotheses effectively in a visual format proved challenging. Consequently, we have decided not to include a figure in this revision.

Original Text (Line 67-70): Using theoretical frameworks of quantitative genetics^{27, 28} and statistical physics²⁹ as our guiding principles, we dissect the relationship of variations across these six levels to elucidate the mechanisms underlying the relationship among variability, variation, microevolution, and macroevolution.

New Text (Line 69-78): We will examine the relationship of variation across these six levels and discuss the role of constraints and congruence hypotheses to explain the paradox of predictability using theoretical predictions of quantitative genetics^{27, 28} and statistical physics²⁹ as our guiding principles. Our basic predictions are that the relationship between standing genetic variation (**G**) and microevolution (**D_{pop}**) will support the constraints hypothesis, while the relationship between variability (**F** or **M**) and macroevolution (**R**) will support the congruence hypothesis. By examining the degree of support for these predictions based on the exponent (i.e., the slope of log-log regression) and the coefficient of determination (R^2) of the relationships, we will propose a pluralistic perspective for understanding the relationship between microevolution and macroevolution.

It is not a new idea that macroevolutionary phenomena can be thought of as the average effect of patterns observed along individual lineages – as mentioned by the authors. This has been demonstrated in various ways – as discussed in the introduction of the present manuscript. However, there’s no current mention about interpretation of things like macroevolutionary regression parameters. I think it would be nice to include a brief discussion / mention of papers that explicitly talk about this in the context of phylogenetic evolution e.g. Hansen et al (2012).

It is interesting that the reviewer associates our idea with the “primary optima” proposed by Hansen (1997), which is a recapitulation of “quantum evolution” and “adaptive zone” proposed by Simpson (1944). We did not consider the primary optima when we crafted our explanations for this study, but your comment prompted us to consider the link, and we agree with the reviewer that the congruence mechanism and the adaptive optima have some relationships. Thus, in our revision, we added a new paragraph mentioning the implication of our results for phylogenetic comparative studies of adaptation (lines 232-242).

Given the importance of the scale of variances being measured, I would expect more information about the divergences and distances amongst the sampled populations. There is very limited information about which populations were sampled – and even less so about why or how. This doesn’t need to be extensive, but just a bit of contextual information would be helpful.

Thank you. Now, we add more details about populations information. Sampling points were chosen based on a minimum straight-line distance of 20 km between them. On the other hand, there was no specific reason (e.g., rural or urban) for selecting each sampling location.

Original Text (Line 232-233): *Drosophila simulans* were sampled from different wild populations in Japan, and isofemale lines were established (Extended Data Table 2).

New Text (Line 301-304): We sampled *Drosophila simulans* from multiple wild locations in Japan, each separated by a straight-line distance of more than 20 km between them (Supplementary Fig. 1). Sampling was conducted through sweeping or by collecting fruits that were likely to have eggs laid on them. Isofemale lines were then established from the sampled adult females (Supplementary Table 2).

Supplementary Fig. 2 The location of sampling points.

Minor comments:

L121: constraint or constraints, not constrains

L269: Figure?

L283: No citation for MCMCglmm

L282-300 Refer reader to R code – but these models are beautifully described!

L299-300: Did replicate analyses converge on the same answers?

Thank you so much for your recommendations. Now, we have checked our manuscripts and corrected our paper based on your minor comments. In addition, we add new Supplementary figure about the position of landmarks.

About replicate analyses, we did run multiple replicates in all models except for the one estimating FA. In all models with replicates, they converged to the same answer. However, for the model that tries to estimate FA covariance matrix, the model was extremely slow and many attempts of running the model with the specified generation did not finish, because the resampling appears to have failed at certain points, and they did not proceed further. We do, however, account for the uncertainty arising from a replicate that did run for the full generation, which are shown in Figure 4.

Supplementary Fig. 3 Position of 12 two-dimensional landmarks utilized in this study.

Dear Editors and Reviewers

Thank you very much for reviewing our manuscript once again and for providing valuable feedback. We sincerely appreciate the time and effort you have dedicated to evaluating our work.

The reviewers' comments are presented in bold black, while our responses are provided in plain blue text.

Responses to the Comments by the Reviews:

Reviewer: 1

I'm satisfied that the authors have appropriately revised their manuscript.

We sincerely thank the reviewer for their positive evaluation and are glad to hear that the revisions have addressed the concerns.

Reviewer: 3

As before, I read the manuscript with much interest. I feel that with the authors revisions, they have substantially improved not only the quality but also the readability. They have addressed all of my questions in full. Therefore, I am happy to say that this manuscript is suitable for publication in Communications Biology.

We sincerely thank the reviewer for their thoughtful and encouraging comments. We are pleased to hear that the revisions have improved both the quality and readability of the manuscript, and that all concerns have been fully addressed.

Reviewer: 4

I think that the authors have thoroughly and thoughtfully addressed the reviewers' comments in this revised version of the manuscript. I have carefully examined both, the revised manuscript and the detailed rebuttal letter, and I find that the authors have responded appropriately and effectively to the concerns raised during the first round of review.

In particular:

- Respect to the interpretation of fluctuating asymmetry (F), I believe that the authors acknowledge the potential confounding effects of minor genetic and environmental variation and clarify their assumptions and methods. They support

their position with references to relevant empirical and theoretical studies and incorporate these clarifications directly into the text. Their response seems balanced, transparent, and in line with current standards in evolutionary quantitative genetics.

- Regarding the Reviewer 2's more extensive critique, I think that the authors significantly improved the clarity of the conceptual framework, especially regarding scaling exponents and the contrasting predictions of the hypotheses they tested (i.e., constraint and congruence hypotheses). They expanded their discussion of the theoretical basis (including Kaneko & Furusawa's model), clarified the origins and estimation methods for all covariance matrices, and included a new summary table that greatly improves readability. Also, methodological details were added or clarified throughout the manuscript and supplemental material. Finally, it seems that the authors addressed the concerns about allometry and directional asymmetry in a proper manner.

Overall, I may say that the authors have strengthened both the scientific rigor and the accessibility of the manuscript. I think that the changes made through the revision process enhanced the manuscript's value which seems a significant contribution to theoretical discussions on the relationship between micro- and macroevolutionary processes.

We sincerely thank the reviewer for taking the time to carefully read our revised manuscript and rebuttal letter, especially as a newly assigned reviewer. We truly appreciate the thorough and thoughtful evaluation.

We are grateful for the positive assessment of our clarifications regarding fluctuating asymmetry (F), the conceptual framework, and methodological details. We are pleased to hear that the revisions have strengthened both the scientific rigor and accessibility of the manuscript, and we are encouraged by your view that our work makes a meaningful contribution to the theoretical discussion on micro- and macroevolutionary processes.

Minor issues

I recommend reading through the manuscript carefully to correct minor grammatical issues throughout the text. Below, I mention some specific examples to illustrate the types of corrections needed.

Thank you for pointing this out. We have carefully reviewed the entire manuscript and made the necessary grammatical corrections.

- Lines 83-84: "First, based on the quantitative genetic model of Lande proposed

the following relationship". I believe this sentence should be rewritten, for example, as follows: "First, based on the quantitative genetic model of Lande, we propose the following relationship".

New Text (Line 83-84): First, the quantitative genetic model of Lande²⁷ proposed the following relationship:

- Line 165: "while such causal relationship should become progressively decay". Again, I think this sentence should be rewritten, for example, as follows: "should progressively decay" or "should become progressively weaker".

New Text (Line 168): while such causal relationship should become progressively weaker

- Line 337: "and all wings were obtained only from female" should be "...from females".

New Text (Line 341-342): all wings were obtained only from females.

- Lines 387-388: "a more formal estimates of the rate of macroevolution than the raw divergence" should probably be "a more formal estimate of the rate..."

New Text (Line 392): a more formal estimate

- Table 1: "Drosophila Simulans" should be *D. simulans*.

- Lines 613-614: "photo" should be "photos".

- Line 617: "It includes both sides of wing and replications" should be rewritten, for example, as follows: "It includes both wings and replicated measurements".

We have now revised Table 1 and the corresponding annotation text.

Lines 132-136: At this point in the manuscript, it may be helpful to briefly clarify two things:

- Why M was estimated using *D. melanogaster* while F was estimated using *D. simulans*?

- Why two estimates of M were used (i.e., M_{hom} and M_{het})?

Even a concise explanation (acknowledging that more details may follow in the Methods or Supplementary Materials) might help readers to better understand the rationale for these choices, which are central to the comparisons being made.

Thank you for your helpful suggestions. We have added explanations to the

Supplementary Information clarifying why **M** was estimated using *D. melanogaster* while **F** was estimated using *D. simulans*, and why two distinct estimates of **M** were used in this study.

New Text (Line 136-138): The rationale for not estimating **M** in *D. simulans* and for using two types of **M** is provided in the Supplementary Note 1.

Lines 155-158: As with the earlier mention of M and F being estimated from different species almost without explanation, it would be helpful to briefly clarify here what is meant by “our dataset” and, perhaps, how it relates to the other data sources mentioned (e.g., D. simulans populations, the DrosoWing Project, and Houle et al.'s dataset). This would help readers understand the structure and origin of the data used for estimating divergence at different evolutionary scales.

Thank you for your suggestions. We have added more details about our datasets.

New Text (Line 159-160): our dataset, which includes the images used to estimate **G**, **E**, and **D**_{pop}, as well as wing photos of *D. lutescens*,